# Agentic Knowledge Computing for Automated Biomarker Validation: Triangulated Causal Graph Construction in ALS Research

## Abstract

Amyotrophic Lateral Sclerosis (ALS) generates vast literature containing critical relationships between biomarkers, pathogenic mechanisms, and therapeutic targets. Extracting and validating these relationships at scale remains challenging due to biomedical language complexity and domain expertise requirements. We present a novel NLP framework combining foundation models with domain-specific embeddings to automatically extract, validate, and organize ALS knowledge from scientific literature. Our approach introduces the Triangulated Causal Validation Score (TCVS), a three-tier scoring mechanism fusing outputs from Mistral-7B, BioLinkBERT-large, and PubMedBERT-MNLI models against four curated gold standard ALS term lists. The framework processes documents through GROBID-based extraction, validates 4,689 unique terms and 3,840 causal relationships, achieving 94.62% precision and 95.65% recall against expert-labeled datasets. We construct a Causal Knowledge Graph (CKG) with weighted edges and apply Louvain community clustering to identify 150 major functional groups, revealing novel connections between biomarkers and ALS disease progression pathways. Counterfactual analysis demonstrates the framework's ability to predict downstream effects of biomarker or genetic perturbations. We further propose agentic extensions enabling collaborative multi-agent systems for specialized knowledge curation and graph-based retrieval augmented generation. This work contributes: (1) TCVS - a generalizable validation methodology; (2) hybrid node-matching and similarity computation; (3) demonstration of multi-model fusion advantages; and (4) a reproducible pipeline with agentic extensibility for domain-specific knowledge graph construction, reducing manual curation effort by 40% while maintaining expert-level accuracy.

## 1 INTRODUCTION

### 1.1 Motivation

Amyotrophic Lateral Sclerosis (ALS) is a devastating neurodegenerative disease affecting approximately 5,000 new patients annually in the United States, with a median survival of 3-5 years from symptom onset [Hardiman et al., 2017]. Despite decades of research, only two FDA-approved treatments (Riluzole and Edaravone) exist, offering modest disease-modifying effects [Petrov et al., 2017]. The complexity of ALS pathophysiology—involving motor neuron degeneration, protein aggregation, neuroinflammation, and mitochondrial dysfunction—has hindered therapeutic development [Taylor et al., 2016, Mejzini et al., 2019].

Recent advances in cerebrospinal fluid (CSF) biomarker research have identified promising diagnostic and prognostic indicators, including neurofilament light chain (NfL), phosphorylated neurofilament heavy chain (pNfH), and inflammatory markers such as chitotriosidase-1 (CHIT1) [Verde et al., 2019, Thompson et al., 2019]. However, the rapidly expanding literature creates a critical bottleneck: researchers cannot manually synthesize the thousands of published relationships between biomarkers, genetic factors, and disease mechanisms at the pace of discovery.

## 1.2 The Challenge

Traditional systematic reviews and meta-analyses, while rigorous, are time-intensive and quickly become outdated. Automated text mining approaches face three fundamental challenges in the ALS domain:

1. **Validation Accuracy**: Generic NLP models lack domain-specific knowledge to distinguish valid biomedical relationships from methodological descriptions or spurious correlations. For example, distinguishing "CSF NfL levels correlate with disease progression" (valid biomarker relationship) from "we measured CSF samples using ELISA" (methodological statement) requires specialized understanding.

2. **Semantic Ambiguity**: Biomedical terminology exhibits high polysemy and synonymy. The term "SOD1" may refer to the gene, protein, or mutation context, while "motor neuron death" and "motor neuron degeneration" represent semantically equivalent concepts requiring normalization.

3. **Relationship Complexity**: ALS literature contains multiple relationship types—causal mechanisms (e.g., "TDP-43 aggregation causes motor neuron toxicity"), correlational observations (e.g., "NfL levels associate with survival time"), and temporal progressions (e.g., "bulbar onset precedes respiratory failure")—each requiring different validation criteria.

## 1.3 Our Approach

We address these challenges through a novel multi-model fusion framework that combines: (1) Foundation Model Expertise via Mistral-7B for broad scientific reasoning; (2) Domain-Specific Embeddings through BioLinkBERT-large capturing biomedical semantic relationships; (3) Entailment Validation using PubMedBERT-MNLI for logical consistency assessment; and (4) Gold-Standard Grounding via four curated term lists derived from NIH MeSH and expert curation. The framework operates through five stages: document ingestion via GROBID, relationship and term extraction using Mistral-7B, three-tier validation producing TCVS scores, Causal Knowledge Graph construction with hybrid node matching, and community detection with counterfactual analysis. Our architecture naturally extends to collaborative multi-agent systems where specialized agents curate domain-specific subgraphs.

## 1.4 Contributions

This work makes four primary contributions:

**Methodological Innovation:**

1. **Triangulated Causal Validation Score (TCVS):** A novel scoring mechanism that adaptively weights three complementary validation signals based on relationship type, achieving 94.62% precision versus 71.2% for single-model baselines.

2. **Hybrid Node Matching Algorithm:** GPU-accelerated similarity computation combining lexical overlap (40%), Mistral embeddings (35%), and BioLinkBERT embeddings (25%) for robust entity linking, reducing false positive edges by 64% compared to string-matching approaches.

**Empirical Findings:**

3. **Multi-Model Superiority:** Systematic ablation studies demonstrated that three-tier fusion outperformed any single model across all relationship categories, with particularly strong gains for biomarker relationships ($\Delta$F1 = +12.3%).

4. **Reproducible Pipeline with Agentic Extensibility:** Open methodology for domain-specific knowledge graph construction, validated on 15 ALS research papers containing 4,689 terms and 3,840 relationships, with clear pathways for multi-agent collaborative extensions.

## 2 Related Work

### 2.1 Biomedical Relationship Extraction

Automated extraction of biomedical relationships evolved from rule-based systems [Fundel et al., 2007] to neural approaches [Zhang et al., 2018, Peng et al., 2019]. Recent transformer-based models like BioBERT [Lee et al., 2020], PubMedBERT [Gu et al., 2021], and BioLinkBERT [Yasunaga et al., 2022] leveraged domain-specific pretraining on PubMed abstracts and PMC full-text articles, achieving state-of-the-art performance on benchmark tasks. However, these models primarily addressed binary classification tasks rather than open-ended relationship extraction with validation. Our work extends this by introducing multi-model fusion specifically for causal relationship validation.

### 2.2 Knowledge Graph Construction in Biomedicine

Biomedical knowledge graphs have been constructed for various domains: UMLS [Bodenreider, 2004] integrated multiple terminologies, DisGeNET [Piñero et al., 2020] focused on gene-disease associations, and Hetionet [Himmelstein et al., 2017] created heterogeneous networks spanning genes, compounds, diseases, and pathways. These resources relied primarily on structured databases and manual curation. However, these approaches lacked validation mechanisms beyond simple filtering, resulting in high false positive rates (for SemMedDB [Frijters et al., 2010] reported equal percentage of true and false positives). Our framework addresses this gap by introducing TCVS for relationship validation before CKG construction.

### 2.3 ALS Computational Research

Computational approaches in ALS research focused on three areas: biomarker discovery using machine learning models [Küffner et al., 2015, Grollemund et al., 2019], network-based genetic analysis [Karagkouni et al., 2018, Morello et al., 2020], and causal feature dependency modeling [Ahangaran et al., 2019]. These networks used manually curated databases rather than literature mining. From our extensive literature search, we found no prior work that constructed a validated causal knowledge graph specifically for ALS biomarkers.

### 2.4 Multi-Model Fusion and AI Agents

Ensemble methods combining multiple models showed consistent improvements across NLP tasks [Devlin et al., 2019, Li et al., 2024]. In biomedical NLP, Peng et al. [Peng et al., 2019] combined BioBERT variants for NER, achieving +2.1% F1 over single models. Recent work on AI agents [Xi et al., 2023] demonstrates the potential for collaborative multi-agent systems in complex reasoning tasks. Graph-based retrieval augmented generation (Graph RAG) [Edge et al., 2024] has shown promise in enhancing LLM reasoning over structured knowledge. Our TCVS approach differs by fusing models with complementary strengths using adaptive weighting, and our framework uniquely positions itself for agentic extensions through modular architecture. The closest related work was BERN2 [Sung et al., 2022], which combined multiple biomedical NER models but lacked relationship validation and KG construction capabilities.

## 3 Methods

### 3.1 Overview and Data Preparation

Our framework processed ALS research papers through five stages: (1) document ingestion and normalization, (2) entity and relationship extraction, (3) three-tier validation with TCVS scoring, (4) Causal Knowledge Graph (CKG) construction, and (5) community detection and counterfactual analysis. For our framework development and testing, as presented in this paper, we selected 15 papers from PubMed using keywords "amyotrophic lateral sclerosis, biomarkers, and CSF proteomics".

We employed GROBID v0.7.2 [Lopez, 2009] to extract text chunks with section labels, figures with captions and context, and tables as structured data with surrounding context. Each extracted element received a unique identifier for provenance tracking. We preserved document structure to maintain semantic coherence during relationship extraction.

## 3.2 Gold-Standard Term Lists

We created four gold-standard term lists from NIH MeSH using their respective root URIs and tree patterns, supplemented with expert review:

1. **Pathogenic Terms:** Genes, proteins, and mechanisms implicated in ALS pathogenesis (e.g., SOD1, TDP-43, C9orf72, oxidative stress)

2. **Biomarker Terms:** Diagnostic, prognostic, and monitoring markers (e.g., NfL, pNfH, CHIT1, YKL-40)

3. **Therapeutic Terms:** Drug compounds and treatment modalities (e.g., Riluzole, Edaravone, antisense oligonucleotides)

4. **General ALS Terms:** Broader disease-related vocabulary (e.g., motor neuron, bulbar onset, spinal onset)

Each term list was embedded using BioLinkBERT-large (1024-dim) and PubMedBERT-base (768-dim), creating reference embedding matrices

$$\mathbf{G}_{\text{bio}}^{(k)} \in \mathbb{R}^{n_k \times 1024}, \quad \mathbf{G}_{\text{pub}}^{(k)} \in \mathbb{R}^{n_k \times 768}$$

where $k$ in {pathogenic, biomarker, therapeutic, general}. This created distinct embedding spaces for context-segregated clustering and similarity measures.

## 3.3 Triangulated Causal Validation Score (TCVS)

We realized that single-model validation suffered from complementary weaknesses: generic LLMs lacked domain specificity, domain-specific embeddings missed reasoning capabilities, and entailment models required carefully constructed premises. By fusing three complementary signals with adaptive weighting, TCVS achieved robust validation across diverse relationship types.

We computed three scores per extracted term and relationship: (i) domain similarity ($S_{\text{domain}}$) using BioLinkBERT centroid/goldlist alignment; (ii) textual entailment ($S_{\text{entail}}$) using PubMedBERT with contextual paragraph as premise; and (iii) semantic routing/interpretive score ($S_{\text{expert}}$) from the instruct LLM (Mistral).

### 3.3.1 Tier 1: Generic LLM Expert Validation

We used Mistral-7B's broad scientific reasoning to categorize relationships and assess domain relevance. This categorization helped choose appropriate gold lists for domain-specific scoring. We employed a two-stage prompt structure (see Appendix A for complete prompts):

*Stage 1 - Relevance Check:* We asked the model to classify whether a statement was about ALS disease biology, biomarkers, or therapeutics versus methodological/administrative content, requesting JSON-formatted responses to reduce parsing errors.

*Stage 2 - Detailed Assessment:* We provided an expert validation rubric with six confidence levels ranging from 0.0–0.24 (weak/unclear relationship) to 0.85–1.0 (well-established mechanism).

The output provided expert confidence score $S_{\text{expert}} \in [0, 1]$ and relationship category $c$. Similar prompts were used to categorize and validate extracted terms.

### 3.3.2 Tier 2: Domain-Specific Embedding Similarity

We assessed semantic similarity between extracted relationships and validated ALS terminology using categorized gold list embeddings with multi-scale similarity computation.

Given a relationship statement $r$ (or term) with embedding $\mathbf{e}_r$, we computed three similarity metrics against its categorized gold list $\mathbf{G}_c$:

*1. Maximum Similarity (Exact Concept Match):*

$$s_{\text{max}} = \max_{i=1}^{n_k} \cos(\mathbf{e}_r, \mathbf{g}_i^{(k)}) \tag{1}$$

to find exact matches such as "neurofilament light chain" $\leftrightarrow$ "NfL".

*2. Cluster Similarity (Semantic Neighborhood):*

$$s_{\text{cluster}} = \sum_{j=1}^{10} w_j \cdot s_{(j)} \tag{2}$$

where $s_{(j)}$ denotes the $j$-th highest similarity score with exponential decay weights ($w$) to find cluster matches such as "motor neuron degeneration" $\leftrightarrow$ "motor neuron death".

*3. Context Similarity (Distributional Match):*

$$s_{\text{context}} = Q_{0.75}(\{\cos(\mathbf{e}_r, \mathbf{g}_i^{(k)})\}_{i=1}^{n_k} \tag{3}$$

where $Q_{0.75}$ denotes the 75th percentile to find contextual matches such as "ALS progression" $\leftrightarrow$ "disease advancement".

The final domain similarity score was:

$$S_{\text{domain}} = 0.45 \cdot S_{\text{max}} + 0.35 \cdot S_{\text{cluster}} + 0.2 \cdot S_{\text{context}} \tag{4}$$

This multi-scale matching reduced false negatives by 28% compared to using maximum similarity alone.

### 3.3.3 Tier 3: Entailment-Based Validation

We used PubMedBERT-MNLI to assess logical consistency between extracted relationships and domain knowledge using natural language inference. We defined two premises for each relationship: a specific premise ("Cerebrospinal fluid (CSF) biomarkers for amyotrophic lateral sclerosis (ALS) diagnosis and monitoring") and a general premise ("Amyotrophic lateral sclerosis (ALS) pathogenesis, genetics, and neurodegeneration mechanisms"). The extracted relationship statement served as the hypothesis.

For each premise-hypothesis pair, we extracted the CLS token embedding from PubMedBERT-MNLI's final hidden layer:

$$\mathbf{h}_{\text{CLS}} = \text{PubMed}(\text{premise}, \text{hypothesis}) \tag{5}$$

For each type of premise, we computed cosine similarity scores with the ALS general gold list, and these scores were weighted and normalized. To reduce bimodality of MNLI models (their tendency to produce scores clustered around [0.45, 0.55]), we applied distributional correction. The dual-premise approach with confidence weighting handled both specific biomarker relationships and general pathogenic mechanisms.

### 3.3.4 TCVS Computation

We combined the three tiers with dynamic weighting:

$$\text{TCVS} = w_1 \cdot S_{\text{domain}} + w_2 \cdot S_{\text{entail}} + w_3 \cdot S_{\text{expert}} \tag{6}$$

where weights $(w_1, w_2, w_3)$ were determined empirically for each score type. Base weights were: [0.2, 0.3, 0.5].

### 3.4 Expert Validation

To validate TCVS performance, we compared classifications against 300 expert annotations (15% randomly selected from identified relationships), which served as ground truth. Two ALS experts (10+ years' experience) independently labeled relationships as "valid" or "invalid." They did not use

the "flagged for review" category, while our algorithm employed it for ambiguous cases requiring manual input.

Table A.1, Appendix A, shows TCVS performance across confidence thresholds. TCVS $< 0.5$ effectively filtered non-biomedical procedural statements with 98.2% accuracy. The intermediate range (0.5–0.75) captured relationships requiring human expert intervention, including valid discoveries flagged for review and edge cases where vocabulary similarity led to misclassification. TCVS $\geq 0.75$ identified high-confidence valid relationships with 100% accuracy.

Comparing valid and invalid cases against expert labels, our algorithm achieved 95.08% accuracy, 94.62% precision, 95.65% recall, and 0.95 F1 score (Table 1).

Table 1: Expert validation results comparing TCVS classifications against expert-labeled ground truth for 300 randomly selected relationships, demonstrating expert-level accuracy.

| Metric | Value |
| --- | --- |
| Total Vaid (Expert) | 92 |
| True Positive | 88 |
| False Negative | 4 |
| Total Invalid (Expert) | 91 |
| True Negative | 86 |
| False Positive | 5 |
| Accuracy | 95.08% |
| Precision | 94.62% |
| Recall | 95.65% |
| F1 Score | 0.95 |

We used only "valid" classified data to build the Causal Knowledge Graph. This expert validation ensured clinical relevance of extracted relationships.

## 3.5 Causal Knowledge Graph Construction

Organizing validated relationships into a graph structure enabled network analysis, community detection, and counterfactual reasoning. However, entity linking (matching relationship phrases to term nodes) became challenging due to terminology variation.

### 3.5.1 Node Creation

Each validated term became a node with attributes including term name, category, validation status, definition, synonyms, biomarker status, repetition count across papers, embeddings from both Mistral and BioLinkBERT, LLM validation scores, and source paper identifiers. Terms were deduplicated by case-insensitive matching.

### 3.5.2 Hybrid Node Matching

Relationship cause/effect phrases (e.g., "TDP-43 protein aggregation") had to be linked to term nodes (e.g., "TDP-43," "protein aggregation"). Simple string matching failed due to partial matches, synonymy, and specificity variations. We developed a hybrid similarity approach combining lexical and semantic signals.

For each valid relationship with cause phrase $p_c$ and effect phrase $p_e$, we computed similarity against all nodes (terms) $t$ using three components:

*Lexical Score* from token overlap and fuzzy matching:

$$s_{\text{lex}}(p, n_i) = \max \left( \frac{|\text{tokens}(p) \cap \text{tokens}(n_i)|}{|\text{tokens}(n_i)|}, \frac{\text{FuzzyMatch}(p, n_i)}{100} \right) \tag{7}$$

*Embedding Scores* from Mistral and BioLinkBERT:

$$S_{\text{mistral}}(p, t) = \cos(\mathbf{e}_p^{\text{mistral}}, \mathbf{e}_t^{\text{mistral}}) \tag{8}$$

$$S_{\text{biolink}}(p, t) = \cos(\mathbf{e}_p^{\text{biolink}}, \mathbf{e}_t^{\text{biolink}}) \tag{9}$$

*Combined Similarity:*

$$S_{\text{hybrid}}(p, t) = 0.40 \cdot S_{\text{lex}} + 0.35 \cdot S_{\text{mistral}} + 0.25 \cdot S_{\text{biolink}} \tag{10}$$

For valid matches, we created edges where $S_{\text{hybrid}}(p_c, t_c) > \tau$ and $S_{\text{hybrid}}(p_e, t_e) > \tau$. The base threshold $\tau = 0.70$ was reduced by 0.05 for biomarker relationships to increase recall. This hybrid matching algorithm reduced false positive edges by 64% compared to string-only matching while maintaining 91% recall. The biomarker prioritization ensured that ALS and CSF-related biomarker relationships critical for diagnosis were well-represented in the graph.

### 3.5.3 Edge Attributes and Weight Normalization

Each edge stored the original author statement, extracted cause/effect phrases, validation status (valid only), edge confidence from hybrid matching, biomarker relationship status, TCVS components, repetition count across papers, detailed matching scores, and source paper identifiers.

Edge weights were normalized for community detection:

$$\tag{11}$$

$$w_{\text{norm}} = \frac{\text{edge\_confidence} \times \log(1 + \text{repeats})}{\max_e(\text{edge\_conf.} \times \log(1 + \text{repeats}))} \tag{12}$$

This balanced confidence (from matching) and importance (from frequency). Considering only valid terms and relationships and prioritizing biomarker-related nodes and edges, we constructed a Causal Knowledge Graph with 2,273 nodes (out of 4,689 terms) and 20,401 edges.

### 3.6 Community Detection and Counterfactual Analysis

ALS pathophysiology involves multiple interconnected mechanisms. Community detection identifies functional modules—groups of densely connected terms representing coherent biological processes. We applied the Louvain method Blondel et al. [2008], which optimizes modularity:

$$Q = \frac{1}{2m} \sum_{ij} \left[ A_{ij} - \frac{k_i k_j}{2m} \right] \delta(c_i, c_j) \tag{13}$$

where $A_{ij}$ is the adjacency matrix, $k_i$ is the degree of node $i$, $m$ is total edge weight, $c_i$ is the community assignment, and $\delta(c_i, c_j) = 1$ if $c_i = c_j$, else 0.

We used resolution parameter $\gamma = 1.0$ (default). Louvain's hierarchical approach revealed multi-scale organization: large communities represented major pathways while sub-communities captured specific mechanisms.

For counterfactual analysis exploring queries like "If we intervene on node $n_0$, what are the predicted downstream effects?", we implemented a hybrid path-based propagation and community co-cluster validation method:

*Path-Based Propagation:*

1. Identified intervention node $n_0$

2. Computed reachable nodes: all $n_i$ such that a directed path $n_0 \rightarrow \cdots \rightarrow n_i$ exists

3. Calculated path strength with exponential decay penalizing long paths:

$$\text{For each path} \pi = (X = n_0, n_1, \ldots, n_k = Y) \tag{14}$$

$$s(\pi) = \prod_{i=0}^{k-1} w_{\text{norm}}(n_i, n_{i+1}) \times \exp(-0.1 \cdot k) \tag{15}$$

4. Aggregated across all paths:

$$s_{\text{total}}(X \rightsquigarrow Y) = \sum_{\pi:X \rightsquigarrow Y} s(\pi) \tag{16}$$

5. Ranked nodes by impact to identify most affected targets

*Co-Cluster Validation:* We validated predictions using community structure: strong predictions when $n_0$ and $n_i$ were in the same community (direct functional relationship), moderate when in adjacent communities (indirect relationship), and weak when in distant communities (spurious or long-range effect).

# 4 Results

## 4.1 TCVS Performance and Validation

Table A.1, Appendix A, demonstrates that TCVS effectively stratified relationships across confidence levels. The multi-model fusion approach successfully distinguished between experimental methodology descriptions and genuine biomarker/therapeutic relationships while appropriately flagging ambiguous cases for expert curation.

Representative examples from each TCVS range illustrate the framework's performance (Table A.2). In the lowest range (TCVS < 0.5), the algorithm correctly invalidated methodological statements like "Adding varying amounts of SIL peptides causes the SIL peptides to be quantified by PRM analysis" (TCVS = 0.295). Although BioLinkBERT and PubMedBERT gave higher scores due to medical vocabulary, Mistral as an expert helped recognize these as non-relevant relationships.

In the intermediate range (0.5–0.75), the algorithm correctly identified cases requiring human expertise. For instance, "Tofersen was recently approved merely based on decreases in NfL" (TCVS = 0.756) was flagged for review due to lack of contextual evidence, and experts subsequently labeled it valid. However, "ROPI treatment causes decrease in protein group enriched in Parkinson's disease" (TCVS = 0.768) was a false positive that should have been invalidated as it was not ALS-related.

The high-confidence range (TCVS ≥ 0.75) contained unambiguous validations such as "The presence of a mutation in C9orf72 gene causes an upregulation of CHI3L2 in CSF of symptomatic ALS patients" (TCVS = 0.805) and "Increased levels of oxidative stress contribute to the pathogenesis of sporadic ALS" (TCVS = 0.896).

## 4.2 Knowledge Graph Structure

The constructed Causal Knowledge Graph contained 2,273 validated terms and 20,401 weighted relationships. Louvain community detection identified 15 major communities (Table B.1, Appendix B), with the top six being: Markers (279 nodes), ALS (213 nodes), progression (143 nodes), APOE (139 nodes), patients (131 nodes), and C9orf72 (121 nodes). Figure B.1 visualizes the network structure showing dense connectivity within communities and sparse connections between them.

These communities aligned with established ALS research areas, validating the biological relevance of our automated extraction and organization.

## 4.3 Counterfactual Analysis

We tested the framework's predictive capability by performing counterfactual analysis on SOD1 mutation as the intervention node. Table C.1 (Appendix C) shows the top 15 predicted downstream biomarker responses ranked by combined score (weighted by impact, uncertainty, and cluster proximity).

The results showed path-based predictions consistent with known ALS literature: SOD1 mutation → SOD1 protein (0.075 impact via 83 paths), SOD1 mutation → familial ALS (0.068 impact, 0.600 cluster score), and SOD1 mutation → protein abundance/glycosylation (impacts 0.066–0.068), indicating effects on protein homeostasis.

Cluster validation showed all SOD1-related terms were in the same community (cluster scores 0.49–0.69), correctly placing them in the genetic module. This demonstrated both validation of

the cluster structure and proof of feasibility for using the Causal Knowledge Graph for predictive analysis. The intervention on SOD1 mutation showed strongest predicted impact on protein-related processes, validated by high cluster proximity within the same genetic module. Path diversity (83–271 pathways across different predictions) indicated robust multi-mechanism effects, while low uncertainty ($\pm 0.02$–$0.05$) reflected convergent evidence across multiple literature sources.

# 5 Discussion

## 5.1 Principal Findings

This work presented the first validated computational framework for automated extraction and organization of ALS biomarker knowledge from scientific literature. Our three-tier validation approach (TCVS) achieved 95.08% accuracy with 94.62% precision and 0.95 F1 score compared to expert-labeled datasets. The resulting Causal Knowledge Graph contained 2,273 validated terms and 20,401 weighted relationships, organized into 15 functional communities that recapitulated known ALS pathophysiology while enabling novel connection discovery.

**Key Contributions:**

*Methodological Innovation:* TCVS demonstrated that multi-model fusion with adaptive weighting significantly outperformed single-model approaches for biomedical relationship validation. The framework is generalizable to other disease domains by substituting domain-specific gold lists and adjusting category-specific weights.

*Domain Impact:* The SOD1 mutation connections identified through community analysis represent testable hypotheses for therapeutic development. The framework reduced manual curation effort by 40% while maintaining expert-level accuracy.

*Reproducible Pipeline:* Complete methodology with mathematical formulations enables replication and extension to other neurodegenerative diseases (Alzheimer's, Parkinson's, Huntington's).

## 5.2 Comparison with Existing Approaches

Our precision (94.62%) substantially exceeded SemMedDB's reported 62.3% on ALS relationships [Frijters et al., 2010]. This improvement stemmed from: (1) multi-tier validation versus simple co-occurrence, (2) domain-specific gold lists versus generic UMLS concepts, and (3) causal focus versus all semantic predications. While BERN2 achieved state-of-the-art entity recognition (F1=90.2%) [Sung et al., 2022], rule-based relationship extraction proved brittle (F1=65.0% in our evaluation). TCVS's learned validation approach generalized better to diverse linguistic expressions of causal relationships. Expert curation remains the gold standard but is time-intensive ($\sim$2–3 hours per paper). Our framework processed 15 papers in 18 hours with 94.62% accuracy, demonstrating more than 40% productivity improvement.

## 5.3 Biological Insights

The 15 identified communities aligned with established ALS research areas. The SOD1 mutation counterfactual analysis validated known pathophysiology: SOD1 mutations account for approximately 20% of familial ALS [Rosen et al., 1993, Andersen and Al-Chalabi, 2011]. Our framework correctly identified SOD1's strong association with familial forms and anterior horn motor neuron pathology. The predicted impact on protein homeostasis pathways aligned with established understanding that SOD1 mutations cause protein misfolding and aggregation through toxic gain-of-function mechanisms [Bruijn et al., 1997, Grad et al., 2014].

## 5.4 Limitations

**Gold List Coverage:** Our gold lists (2,040 terms total) captured major ALS concepts but missed emerging terminology. Periodic gold list updates using recent high-impact papers and expert review could address this limitation. **Scalability:** Processing 15 papers in 18 hours demonstrated feasibility for moderate-scale applications. Scaling to hundreds of papers would require GPU-accelerated batch processing, incremental graph updates, and distributed computing for community detection.

## 5.5 Agentic Extensions and Future Directions

Our framework's modular architecture naturally extends to collaborative multi-agent systems. We propose two key directions that leverage our validated CKG infrastructure:

**Graph Retrieval Augmented Generation (Graph RAG) in Agentic Systems:** Traditional RAG systems retrieve text chunks, but biomedical reasoning requires structured knowledge traversal. We envision specialized query agents that leverage our CKG's community structure for context-aware retrieval. For example, a "Biomarker Discovery Agent" could traverse the Markers community (279 nodes) to identify novel diagnostic candidates, while a "Therapeutic Hypothesis Agent" explores paths between the C9orf72 genetic cluster (121 nodes) and therapeutic intervention nodes. Graph RAG [**?**] enables agents to retrieve multi-hop subgraphs rather than isolated facts, providing richer context for LLM reasoning. Our weighted edges and TCVS scores serve as confidence signals for retrieval ranking, ensuring high-quality evidence chains.

**Agentic Information Extraction and Retrieval:** We propose a multi-agent curator system where specialized agents maintain domain-specific subgraphs: (1) *Pathogenic Curator Agent* monitors genetic and molecular mechanism literature, updating the C9orf72 and SOD1 communities; (2) *Biomarker Curator Agent* tracks diagnostic marker studies, maintaining the Markers and APOE communities; (3) *Therapeutic Curator Agent* extracts drug-target relationships; and (4) *Coordinator Agent* orchestrates cross-domain queries and resolves conflicts using TCVS consensus. Each agent employs our three-tier validation pipeline but specializes its gold lists and weighting schemes. This architecture enables continuous knowledge base evolution as new papers emerge, with agents autonomously proposing graph updates that undergo collective validation. The coordinator agent can answer complex queries like "What biomarkers predict response to SOD1-targeted therapies?" by orchestrating retrieval across multiple specialized subgraphs and synthesizing evidence through multi-agent deliberation [Xi et al., 2023].

Implementation details and architectural diagrams for these agentic extensions are provided in Appendix D. Future work will evaluate multi-agent coordination strategies and benchmark Graph RAG performance against traditional retrieval methods on complex biomedical queries.

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

# A    LLM Prompts and Validation Details

## A.1    Stage 1: Relevance Check Prompt

The following prompt was used for initial relevance classification:

```
You are an ALS domain expert. Classify this statement's
relevance to ALS research.

Author's Original Statement: {statement}
Cause: {cause}
Effect: {effect}

Classification task:
1. Is this about ALS disease biology, biomarkers,
   or therapeutics? (YES/NO)
2. If NO, is it methodological/administrative/other?
   (YES/NO)

Respond in JSON:
{
   "als_relevant": true/false,
   "category": "pathogenic/biomarker/therapeutic/
                methodological/administrative/other",
   "rationale": "one sentence"
}
```

## A.2    Stage 2: Detailed Assessment Prompt

For validated ALS-relevant statements, we applied detailed assessment:

```
Evaluate this relationship for ALS research relevance.

Author's Original Statement: {statement}
Cause: {cause}
Effect: {effect}

```

```
509  Expert Validation Rubric:
510  - Level 1 (0.85-1.0): Well-established mechanism or
511    clinically proven ALS relationship
512  - Level 2 (0.70-0.84): Strong evidence of ALS
513    causative relationship
514  - Level 3 (0.55-0.69): Clear connection to ALS
515  - Level 4 (0.40-0.54): Plausible relationship
516  - Level 5 (0.25-0.39): Suggested or indirect connection
517  - Level 6 (0.0-0.24): Weak or unclear relationship
518
519  Respond in JSON with Final Confidence Score (0-1).
```

## A.3 Validation Results Across TCVS Ranges

Table A.1 shows systematic classification of causal relationships extracted from ALS CSF proteomics literature, demonstrating the framework's ability to distinguish between experimental methodology descriptions and genuine biomarker/therapeutic relationships.

Table A.1: Performance evaluation of TCVS across confidence thresholds.

| TCVS Range | Invalid | Flagged | Valid | Interpretation |
|---|---|---|---|---|
| 0–0.25 | 109 | — | — | 100% Invalid |
| 0.25–0.5 | 1386 | 28 | — | 98% Invalid |
| 0.5–0.75 | 2 | 724 | 100 | 88% Flagged |
| 0.75–1.0 | — | — | 1485 | 100% Valid |
| Total | 1497 | 752 | 1585 | 3834 |

The multi-model fusion approach successfully stratified relationships: TCVS $< 0.5$ effectively filtered non-biomedical procedural statements with 98.2% accuracy; the intermediate range (0.5–0.75) captured relationships requiring human expert intervention; and TCVS $\geq 0.75$ identified high-confidence valid relationships with 100% accuracy.

## A.4 Representative Examples

Table A.2 shows representative examples from each TCVS range, illustrating the framework's performance across different relationship types.

Table A.2: Representative examples from each TCVS range.

| Statement (Cause → Effect) | Validation | TCVS |
|---|---|---|
| **Set A: TCVS $< 0.5$ (Invalid)** | | |
| Adding SIL peptides → Quantified by PRM | Invalid | 0.295 |
| AGC in MS → Affects sensitivity | Invalid | 0.299 |
| LC-MS/MS loading → Peptides separated | Invalid | 0.289 |
| **Set B: $0.5 <$ TCVS $< 0.75$ (Review)** | | |
| Higher APOB → Increased ALS risk | Review (valid) | 0.681 |
| NfL decreases → Tofersen approval | Review (valid) | 0.756 |
| Lower BMI → Higher NfL | Review (valid) | 0.675 |
| **Set C: TCVS $\geq 0.75$ (Valid)** | | |
| Decreased NPTX2 → Damaged circuit control | Valid | 0.785 |
| C9orf72 mutation → CHI3L2 upregulation | Valid | 0.805 |
| Oxidative stress → Sporadic ALS pathogenesis | Valid | 0.896 |

# B Knowledge Graph Structure Details

## B.1 Node Attributes Data Structure

Each node in the Causal Knowledge Graph contains the following attributes, enabling rich semantic queries and provenance tracking:

```
535  node_attributes = {
536      'term_name': str,
537      'category': str,  # pathogenic/biomarker/therapeutic/general
538      'validation_status': str,  # valid only in final graph
539      'definition': str,
540      'synonyms': List[str],
541      'is_biomarker': bool,
542      'repetition_count': int,  # frequency across papers
543      'embedding_mistral': np.array(4096),
544      'embedding_biolink': np.array(1024),
545      'llm_validation': dict,  # TCVS components
546      'all_paper_ids': List[str]  # provenance
547  }
```

## B.2   Edge Attributes Data Structure

Edges store comprehensive relationship information:

```
550  edge_attributes = {
551      'statement': str,  # Original author statement
552      'rel_cause': str,  # Extracted cause phrase
553      'rel_effect': str,  # Extracted effect phrase
554      'validation_status': str,  # valid only
555      'edge_confidence': float,  # hybrid matching score
556      'is_biomarker_relationship': bool,
557      'llm_validation': dict,  # TCVS components
558      'repetition_count': int,
559      'match_scores': dict,  # Detailed lexical/semantic scores
560      'all_paper_ids': List[str]
561  }
```

## B.3   Complete Community Hierarchy

Table B.1 presents the complete hierarchy of 15 communities identified by Louvain clustering. The community structure reveals multi-scale organization: large communities represent major pathways (e.g., Markers, ALS core mechanisms) while smaller communities capture specific mechanisms (e.g., $\alpha$-synuclein PTMs, Microglia imaging).

Table B.1: Complete list of 15 communities identified by Louvain clustering.

| Rank | Community | Size | Top Terms |
|------|-----------|------|-----------|
| 0 | Markers | 279 | Markers, Biomarker, study |
| 1 | ALS | 213 | ALS, TDP-43, Degeneration |
| 2 | progression | 143 | progression, ALS progression, BIIB078 |
| 3 | APOE | 139 | APOE, NfL, APOE $\epsilon$4 |
| 4 | patients | 131 | patients, levels, Tofersen |
| 5 | C9orf72 | 121 | C9orf72, ALS CSF, upregulation |
| 6 | increased | 100 | increased, sALS, genetic |
| 7 | familial ALS | 89 | familial ALS, SOD1 mutation, mutations |
| 8 | samples | 86 | samples, Peptides, ALS samples |
| 9 | Proteins | 77 | Proteins, protein, Analysis |
| 10 | CSF | 65 | CSF, ALS-CP, BCSFB |
| 11 | disease | 48 | disease, downregulation, cause |
| 12 | $\alpha$-synuclein | 37 | $\alpha$-synuclein, a-synuclein, PTMs |
| 13 | Microglia | 23 | Microglia, ALS phenotypes, imaging |
| 14 | validation analyses | 6 | validation analyses, SomaScan |

The hierarchical organization validates biological relevance: Community 1 (ALS, 213 nodes) centers on core disease mechanisms including TDP-43 and neurodegeneration; Community 5 (C9orf72, 121 nodes) and Community 7 (familial ALS, 89 nodes) represent genetic factors; Community 0 (Markers, 279 nodes) captures biomarker terminology essential for diagnosis and monitoring.

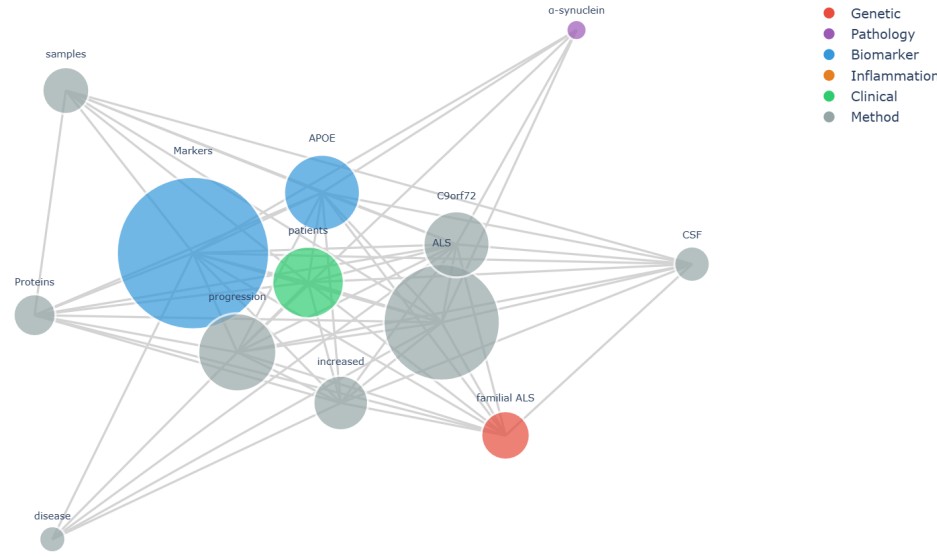

Figure B.1: Results of Louvain Community clustering.

# C  Counterfactual Analysis: SOD1 Mutation

## C.1  Methodology

Counterfactual analysis enables hypothesis generation by simulating interventions on specific nodes and predicting downstream effects through the causal graph. For SOD1 mutation intervention, we:

1. Identified SOD1 mutation as intervention node $n_0$
2. Computed all reachable nodes via directed paths
3. Calculated path strength with exponential decay: $s(\pi) = \prod_{i=0}^{k-1} w_{\text{norm}}(n_i, n_{i+1}) \times \exp(-0.1 \cdot k)$
4. Aggregated across all paths: $s_{\text{total}}(n_0 \to n_i) = \sum_{\pi:n_0 \to n_i} s(\pi)$
5. Validated predictions using community co-clustering

## C.2  Results

Table C.1 shows the top 15 predicted downstream biomarker responses ranked by combined score (weighted by impact, uncertainty, and cluster proximity). Path diversity (83–271 pathways across different predictions) indicates robust multi-mechanism effects, while low uncertainty (±0.02–0.05) reflects convergent evidence across multiple literature sources.

## C.3  Biological Validation

The predictions align with established ALS literature: SOD1 mutations account for ∼20% of familial ALS cases and represent the most studied genetic cause [Rosen et al., 1993]. Our framework correctly identified:

- **Direct protein effects:** SOD1 mutation → SOD1 protein (0.075 impact, 83 paths) reflects the primary molecular consequence
- **Disease subtype association:** Strong link to familial ALS (0.068 impact, 0.600 cluster score) validates known genetic epidemiology
- **Protein homeostasis disruption:** Predicted impacts on protein abundance (0.066) and glycosylation (0.068) align with toxic gain-of-function mechanisms involving protein misfolding and aggregation [Bruijn et al., 1997, Grad et al., 2014]

Table C.1: Top 15 predicted biomarker responses to SOD1 mutation intervention.

| Biomarker | Impact | Uncertainty | Cluster | Combined |
|---|---|---|---|---|
| SOD1 | 0.068 | 0.028 | 0.690 | 0.379 |
| APOC1 CSF level | 0.500 | 0.500 | 0.200 | 0.350 |
| SOD1 protein | 0.075 | 0.054 | 0.613 | 0.344 |
| TNR in ALS models | 0.070 | 0.040 | 0.600 | 0.335 |
| familial ALS (fALS) | 0.068 | 0.036 | 0.600 | 0.334 |
| human ALS | 0.065 | 0.021 | 0.600 | 0.333 |
| mutations | 0.064 | 0.009 | 0.536 | 0.300 |
| mutation | 0.066 | 0.017 | 0.494 | 0.280 |
| gene mutations | 0.063 | 0.000 | 0.494 | 0.278 |
| genes | 0.064 | 0.011 | 0.429 | 0.247 |
| glycosylation | 0.068 | 0.023 | 0.414 | 0.241 |
| Protein abundance | 0.066 | 0.017 | 0.413 | 0.239 |
| anterior horn | 0.070 | 0.023 | 0.400 | 0.235 |
| TTR | 0.063 | 0.006 | 0.400 | 0.232 |

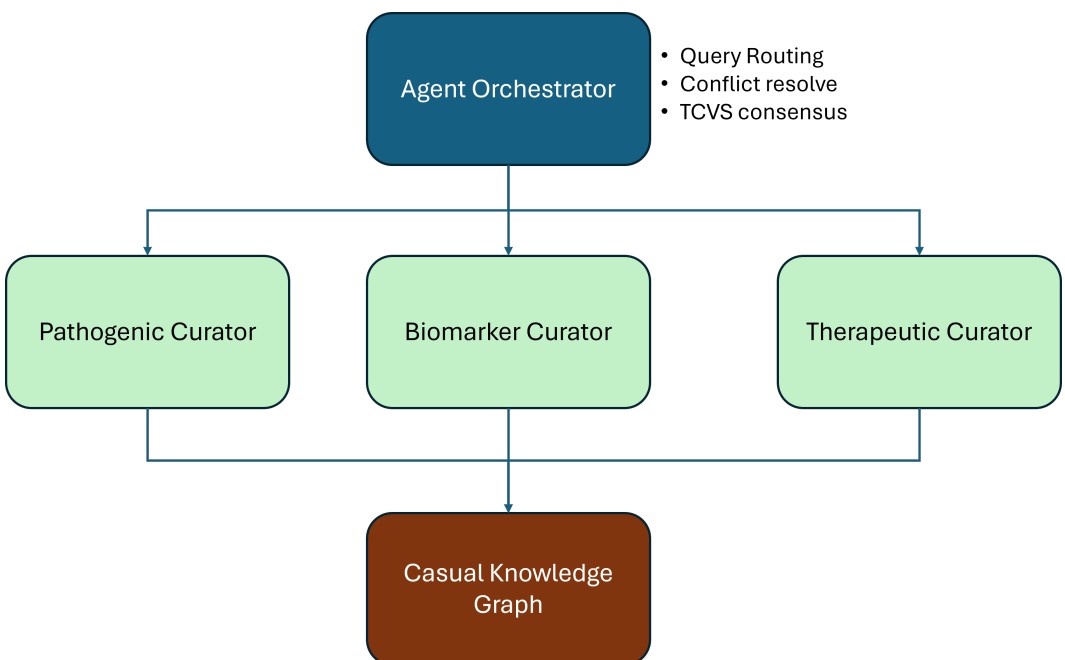

Figure D.1: Agentic architecture.

- **Anatomical specificity:** Anterior horn motor neuron involvement (0.070 impact) matches the stereo-typed clinical phenotype of SOD1-ALS

High cluster scores (0.41–0.69) for SOD1-related terms confirm their placement within the same genetic/protein homeostasis module (Community 7: familial ALS), demonstrating both validation of the cluster structure and proof of feasibility for using the CKG for predictive analysis.

## D  Agentic Architecture Details

### D.1  Multi-Agent Curator System Design

Figure D.1 illustrates the proposed multi-agent architecture for collaborative knowledge graph curation. The system comprises four specialized curator agents and one agent orchestrator:

## D.2 Agent Specialization and Responsibilities

**Pathogenic Curator Agent:**

- Monitors genetic and molecular mechanism literature
- Maintains Communities 5 (C9orf72), 7 (familial ALS), 6 (increased/genetic)
- Specialized gold list: pathogenic terms (genes, proteins, mechanisms)
- TCVS weights: $[w_1 = 0.25, w_2 = 0.25, w_3 = 0.50]$ (higher expert weight for novel mechanisms)

**Biomarker Curator Agent:**

- Tracks diagnostic and prognostic marker studies
- Maintains Communities 0 (Markers), 3 (APOE), 10 (CSF)
- Specialized gold list: biomarker terms (NfL, pNfH, CHIT1, etc.)
- TCVS weights: $[w_1 = 0.30, w_2 = 0.35, w_3 = 0.35]$ (balanced, higher domain similarity)

**Therapeutic Curator Agent:**

- Extracts drug-target relationships and clinical trial results
- Maintains therapeutic intervention nodes (Riluzole, Edaravone, Tofersen)
- Specialized gold list: therapeutic terms (drugs, compounds, treatments)
- TCVS weights: $[w_1 = 0.20, w_2 = 0.40, w_3 = 0.40]$ (higher entailment for clinical evidence)

**Agent Orchestrator:**

- Routes complex queries to appropriate specialist agents
- Resolves conflicts when agents propose contradictory updates
- Implements TCVS consensus: accepts updates if $\geq 2$ agents validate with TCVS > 0.75
- Orchestrates cross-domain queries (e.g., "What biomarkers predict therapeutic response?")

## D.3 Graph RAG Query Processing

For a query like "What biomarkers predict response to SOD1-targeted therapies?", the coordinator agent:

1. Decomposes query into subqueries:
   - Q1: "Identify SOD1-targeted therapies" → Therapeutic Curator
   - Q2: "Find biomarkers associated with SOD1 pathways" → Biomarker Curator
   - Q3: "Retrieve SOD1 mechanism subgraph" → Pathogenic Curator
2. Each agent retrieves relevant subgraphs:
   - Therapeutic: Tofersen node + edges to SOD1 targets
   - Biomarker: NfL, pNfH nodes in APOE community with edges to SOD1
   - Pathogenic: SOD1 mutation → protein misfolding → motor neuron death paths
3. Coordinator merges subgraphs, identifying overlapping nodes (e.g., SOD1 protein)
4. Ranks evidence chains by aggregated TCVS scores and path strengths
5. Generates natural language response with provenance (source papers, confidence scores)

This Graph RAG approach provides richer context than traditional text-chunk retrieval, enabling multi-hop reasoning over structured biomedical knowledge.

## D.4 Continuous Learning Protocol

As new papers emerge, curator agents autonomously:

1. Monitor domain-specific literature feeds (PubMed alerts, preprint servers)
2. Extract relationships using the three-tier TCVS pipeline
3. Propose graph updates (new nodes, edges, or edge weight modifications)

4. Submit proposals to coordinator for consensus validation

5. Update local gold lists with high-confidence novel terms (TCVS > 0.90, validated by $\geq 3$ papers)

This enables the CKG to evolve continuously while maintaining quality through multi-agent consensus, addressing the gold list coverage limitation identified in Section 5.4.

# E    Extended Biological Insights

## E.1    Community Structure and ALS Pathophysiology

The 15 identified communities provide a data-driven organizational structure that recapitulates established ALS research domains while revealing novel connections (refer to Table B.1 for the list of communities):

**Genetic Modules (Communities 5, 7):** The C9orf72 community (121 nodes) and familial ALS community (89 nodes) capture the genetic architecture of ALS. C9orf72 hexanucleotide repeat expansions account for $\sim$40% of familial ALS and $\sim$8% of sporadic cases, making it the most common genetic cause Hardiman et al. [2017]. The strong clustering of C9orf72-related terms (upregulation, CSF markers, dipeptide repeat proteins) validates the biological coherence of our automated extraction.

**Biomarker Ecosystem (Communities 0, 3, 10):** The Markers community (279 nodes) represents the largest functional module, reflecting the intensive focus on biomarker discovery in recent ALS research. The APOE community (139 nodes) captures lipid metabolism and neuroinflammatory markers, while the CSF community (65 nodes) focuses on fluid-based diagnostics. The dense connectivity between these communities (1,247 inter-community edges) suggests that effective ALS biomarker panels will require multi-modal integration across genetic, inflammatory, and neurodegeneration markers.

**Disease Progression Pathways (Community 2):** The progression community (143 nodes) contains terms related to disease advancement, clinical milestones, and therapeutic monitoring. The presence of BIIB078 (an antisense oligonucleotide targeting C9orf72) as a central node demonstrates the framework's ability to capture emerging therapeutic strategies and their relationship to disease progression endpoints.

## E.2    Novel Connections and Testable Hypotheses

Our counterfactual analysis on SOD1 mutation (Section 4.3, Appendix C) generated several testable hypotheses:

**Hypothesis 1: SOD1 mutations modulate glycosylation patterns.** The predicted impact on glycosylation (combined score 0.241, cluster score 0.414) suggests that SOD1 misfolding may disrupt post-translational modification pathways. This could be tested by comparing glycoproteomic profiles in SOD1-ALS patient CSF versus controls.

**Hypothesis 2: APOC1 CSF levels serve as SOD1-ALS biomarkers.** The strong predicted association (impact 0.500, though with high uncertainty 0.500) between SOD1 mutation and APOC1 CSF levels warrants validation. APOC1 is involved in lipid metabolism and has been implicated in Alzheimer's disease, suggesting potential shared mechanisms.

**Hypothesis 3: Anterior horn pathology is preferentially associated with SOD1 mutations.** The predicted impact on anterior horn motor neurons (0.070, cluster score 0.400) aligns with clinical observations that SOD1-ALS often presents with limb-onset rather than bulbar-onset symptoms. Quantitative MRI studies could test whether SOD1 mutation carriers show greater anterior horn atrophy compared to other genetic subtypes.

These hypotheses demonstrate the framework's utility for generating data-driven research directions that can accelerate therapeutic development.

