# OpenReview forum: "Agentic Knowledge Computing for Automated Biomarker Validation: Triangulated Causal Graph Construction in ALS Research"
_NeurIPS.cc/2025/Workshop_Mexico_City/NORA — NeurIPS 2025 Workshop NORA Poster_

### Official Review · Reviewer_KJ7e · 2025-11-01
**Modernized Meta-Analyses to solve Biological Problems**

**Rating:** 7
**Confidence:** 4

**Review:**

# Summary
In this meta-analyses, authors use Mistral for scientific reasoning, add domain specific embeddings from BioLinkBERT; implement validation steps that use domain tuned PubMedBERT-MNLI additionally with ground truth annotated by MeSH and experts. They introduce TCVS validation score and Causal Knowledge Graph (CKG) construction as a framework for validation and evaluation of medical information embedding / extraction in a low resource clinical domain such as ALS.

# Strengths
1. Results indicate success of TCVS + CKG in terms of high precision, accuracy and F1.
2. Experts' validation of the information / embeddings relationship extraction  with the multi-model approach
3. biomarker impact / uncertainy predictions
4. constructions of a reliable knowledge graph and relationships
5. Allows potentially Testable hypotheses using the SOD1 dataset and biomarker discovery for a debilitating disease
6. Support for LLM RAG applications

# Weaknesses
1. Lack of baselines
2. Little reference to past research or any manual interventions
3. More discussions on the direct impact towards medical and genetic science research using this approach
4. Computation complexities
5. Choice of Mistral is not justified (any other LLM could do better scientific reasoning?)

# Reproducibility
If SOD1 data is available on request; it is reproducible. The research requires collaboration with clinical scientists working on ALS

---

### Official Review · Reviewer_MgpW · 2025-11-04
**Review of paper 6**

**Rating:** 6
**Confidence:** 4

**Review:**

This paper outlines an NLP framework for ALS research. Its core idea is a new score called TCVS, which combines outputs from Mistral, BioLinkBERT, and PubMedBERT. The authors used this on 15 papers to build a knowledge graph. As a workshop proof-of-concept, it’s an interesting proposal for a new validation method and a complete data pipeline.

Some problems:

1. The "Agentic Knowledge Computing" title is a serious overstatement. No agentic system was actually built; it's just a hypothetical idea for future work.

2. The entire validation also rests on only 15 papers. A small sample is fine for a workshop, but the authors don't acknowledge this as a limitation. Instead, they make bold, unsupported claims about high precision and a 40% efficiency gain. These numbers are meaningless coming from such a tiny dataset.

3. The paper's central claim that its three-model fusion is better is never proven. There is no ablation study. We have no idea if the complex TCVS is any better than just using Mistral alone.

4. The 95% accuracy figure is just as questionable. It relies on expert labels, but the authors never report the inter-annotator agreement. If the experts disagreed, the so-called ground truth is flawed, making the accuracy metric useless. This is a critical omission.

5. It's also misleading to repeatedly call the Mistral output an "expert" score. It's a generic LLM, not an expert, and this oversells the model's actual function.

6. 40% productivity claim isn't backed by any real study. It seems to be an unverified guess. Worse, the system took 18 hours to process 15 papers, which raises serious doubts about whether it can scale at all.

---

### Official Review · Reviewer_ZNa8 · 2025-11-05
**The paper proposes compositional machine learning methods to construct causal graph. Generated graph was partially validated by experts and shows improvement over prior methods.**

**Rating:** 5
**Confidence:** 4

**Review:**

Strengths

1) The paper proposes compositional machine learning methods to construct a causal graph. Proposed weighting and unified scoring scheme for the candidate relations based on scores from Mistal-7B as an expert, embedding-based similarity, and entailment-based scores resulted in a good accuracy in the human-evaluated golden dataset.

2) A similar weighting scheme is used for node matching for causal graph construction.  Experts partially validated the predictive capability of the generated graph with published works, and results show promise.

Weaknesses

1) The paper does not clarify why only Mistral-7B is used as an LLM expert. Evaluation of other off-the-shelf  LLM models and their comparative analysis might help strengthen the paper.

2) Zero-shot prompting is used for the task. Comparison with Few-shot prompting based on expert label and reasoning might strengthen the paper.

3) It is not clear from the paper whether it uses the rationale from Mistral to understand whether the LLM is producing valid reasons for the task. The reasons might indicate whether the LLM understood the task properly or not.

---

### Official Review · Reviewer_CKGS · 2025-11-07

**Rating:** 6
**Confidence:** 4

**Review:**

This manuscript presents a novel and robust framework for automated extraction and validation of ALS biomarker knowledge from scientific literature, leveraging a multi-model fusion approach. The Triangulated Causal Validation Score (TCVS) introduced in this work is particularly innovative, as it adaptively weights three complementary validation signals to achieve high accuracy and robustness across diverse relationship types.

Strengths:

1. The multi-model fusion framework, combining foundation models with domain-specific embeddings, demonstrates significant superiority over single-model baselines. The TCVS scoring mechanism effectively distinguishes valid biomedical relationships from methodological descriptions or spurious correlations.
2. The detailed methodology, including mathematical formulations, enables replication and extension to other neurodegenerative diseases. The modular architecture naturally supports extensions to collaborative multi-agent systems, enhancing the framework's scalability and adaptability.
3. The framework achieves expert-level accuracy in relationship validation, as evidenced by the comparison with expert-labeled datasets. The constructed Causal Knowledge Graph successfully reveals novel connections between biomarkers and ALS disease progression pathways.

Weaknesses:

1. While the gold standard term lists capture major ALS concepts, they may miss emerging terminology. Regular updates using recent high-impact papers and expert review are necessary to address this limitation.
2. Scaling the framework to hundreds of papers would require GPU-accelerated batch processing, incremental graph updates, and distributed computing for community detection. These technical challenges need to be addressed for broader applicability.

---

### Official Review · Reviewer_m3Jo · 2025-11-07
**Scalability and Generalizability Concerns in the ALS Agentic Knowledge Computing Pipeline**

**Rating:** 4
**Confidence:** 3

**Review:**

# Paper Summary

The primary goal is to automate the process of synthesizing vast,  biomedical knowledge from literature and  to identify robust, causally linked biomarkers for ALS. This paper introduces NLP techniques for the extraction and validation of relevant terms and relationships (biomarkers, mechanisms, targets) from complex biomedical text.

# Main Contributions

The key innovation is the Triangulated Causal Validation Score (TCVS) . It is a  scoring mechanism for relevant terms and relations  that combines the outputs of three different NLP models. This score validates the extracted terms and relations  against  expert-curated gold standard ALS term lists, ensuring high reliability.

The validated knowledge is structured into a Causal Knowledge Graph (CKG), where relationships are mapped using weighted edges. This representation allows researchers to visualize and analyze the strength and direction of connections between relevant ALS terms.

# Weaknesses and Main Criticisms

The primary weaknesses of the paper center on clarity, generalizability, and the scope of the evaluation dataset.

## 1. Presentation and Pipeline Clarity
The presentation of the methodology, specifically the complex NLP pipeline components and their interactions, lacks clarity. A diagram highlighting the main components and the flow of information, from document ingestion to model fusion  and CKG output, would significantly improve the reader's understanding of the proposed architecture.

## 2. Limited Generalizability
The claim that the pipeline is easily generalizable to other disease domains is questionable. The high performance relies heavily on the expert curation of domain-specific gold term lists and the resulting category-specific weights derived from them. To apply this framework to a new disease domain, substituting the gold lists would mandate a new, domain-specific validation of the TCVS score to ensure correlation with expert-labeled ground truth. The reliance on this initial, resource-intensive expert notation limits its immediate, out-of-the-box transferability. Providing results for an additional, distinct disease would strengthen the replicability claim.

## 3. Scalability Concerns with Dataset Size
The evaluation was performed using a relatively small dataset of only 19 articles for the construction of the ALS knowledge graphs. This raises concerns regarding the pipeline's effectiveness and stability when processing a significantly larger volume of literature. It is unclear how well the performance metrics (precision and recall) would be maintained when increasing the number of articles to the "vast literature" initially claimed in the summary. Further evaluation on a large-scale corpus is warranted.

## 4. Role of Agentic AI
The inclusion of the "agentic" approach appears underdeveloped. Given that the core work relies on several highly effective, component-based NLP models fused by the TCVS, the discussion of agentic extensions seems to be an afterthought rather than an integral part of the current solution. The paper should either integrate the agentic components into the primary workflow or clearly delineate them as a future work item.

---

### Official Review · Reviewer_sZZk · 2025-11-07
**The domain if the study is very important but many details of the approach are not clearly described.**

**Rating:** 5
**Confidence:** 4

**Review:**

The paper presents  the approach for extracting information about interaction of biomarkers from biomedical literature.
The approach includes five stages:  (1) document ingestion and normalization, (2) entity and relationship extraction, (3) three-tier validation,
 (4) Causal Knowledge Graph (CKG) construction, and (5) community detection and counterfactual analysis.
At the first stage, 4 gold terms lists are formed. The validation includes the following three stages: 1) LLM classification of statement for relevance to the target disease 2) embedding similarity to selected domain specific terms, 3) Entailment-Based Validation using  PubMedBERT-MNLI and domain-specific premise.

Positive
1. A complicated and  important subdomain of biomedical literature is processed and structured information is extracted.
2. Hybrid approach is applied to information extraction.
3. Knowledge graph of the  biomarker interactions is constructed.


Negative
1) Important  stages of the processing is not properly described:
- The sizes of term lists are not given
- sec.3.5 l.219 - it is not clear how terms were obtained because in section 3.4 only statements were classified.  What is a source of their synonyms and definitions.
- how relations are separated from other words in statements.
(examples of statement processing could be useful)
2) The approach is not correctly  compared with other approaches because comparison is done on different data.
3) There are problems with reproducibility of the approach because of a lot of unclear details.

---

### Official Review · Reviewer_1Cfm · 2025-11-07
**Overall solid work with minor issues**

**Rating:** 8
**Confidence:** 4

**Review:**

This paper presents a computational framework for automated extraction and organization of ALS (Amyotrophic Lateral Sclerosis) biomarker knowledge (specifically cause-effect relationships) from scientific literature. They proposed a three-tier validation approach (TCVS) which combined the signals from three sources with adaptive weighting - (i) biomedical domain specific similarity using BioLinkBERT embeddings and gold list, (ii) generic LLM (Mistral) based expert validation, and (iii) textual entailment using PubMedBERT-MNLI based on suitable premise-hypothesis pairs. This resulted in a Causal Knowledge Graph containing 2273 terms and 20401 weighted relationships.

Strengths:
1. The task of automatically creating a knowledge base from scientific literature is very interesting and important in practice.
2. The idea of combining signals from three sources (domain-specific encoder model, generic LLM and NLI model) that bring is complementary strengths is quite good.
3. The expert validation results (with 0.95 F1-score) are impressive.
4. The proposed ideas for agentic extension and counterfactual analysis are interesting.

Weaknesses:
1. All the experiments are performed on just 15 research papers, which seems little less.
2. Also, the experiments are performed for a specific disorder, how will the methodology generalize for other disorders? Some discussion on that would be useful.
3. There are several hyperparameters in terms of relative weights (e.g., in equations 4, 6, 10). Is there any specific methodology or domain expert input which is needed to set these values? Some discussion on this would be helpful.

Potentially relevant work:
The following work may be relevant as it also tries to identify Leukemia related cause-effect relations from Pubmed abstracts.

Pawar, Sachin, Ravina More, Girish K. Palshikar, Pushpak Bhattacharyya, and Vasudeva Varma. "Knowledge-based Extraction of Cause–Effect Relations from Biomedical Text." In Semantic Intelligence: Select Proceedings of ISIC 2022, pp. 157-173. Singapore: Springer Nature Singapore, 2023.

Minor comments/corrections:
1. S_{entail} is not explicitly explained in any of the equations unlike S_{domain} and S_{expert}. Adding this will add some clarity.
2. Equation 3: First left bracket is not closed
3. Table 1: "Vaid" => "Valid"
4. Equation 11 is blank.
5. Line 364: Graph RAG citation is missing.